# A mitochondrial carrier transports glycolytic intermediates to link cytosolic and mitochondrial glycolysis in the human gut parasite *Blastocystis*

Eva Pyrihová[1,2], Martin S King[1], Alannah C King[1], M Rey Toleco[2], Mark van der Giezen[2,3]*, Edmund RS Kunji[1]*

[1]Medical Research Council Mitochondrial Biology Unit, The Keith Peters Building, Cambridge, United Kingdom; [2]University of Stavanger, Department of Chemistry, Bioscience, and Environmental Engineering, Stavanger, Norway; [3]Research Department Stavanger University Hospital, Stavanger, Norway

**\*For correspondence:**
mark.vandergiezen@uis.no (MvdG);
ek@mrc-mbu.cam.ac.uk (ERSK)

**Competing interest:** The authors declare that no competing interests exist.

**Abstract** Stramenopiles form a clade of diverse eukaryotic organisms, including multicellular algae, the fish and plant pathogenic oomycetes, such as the potato blight *Phytophthora*, and the human intestinal protozoan *Blastocystis*. In most eukaryotes, glycolysis is a strictly cytosolic metabolic pathway that converts glucose to pyruvate, resulting in the production of NADH and ATP (Adenosine triphosphate). In contrast, stramenopiles have a branched glycolysis in which the enzymes of the pay-off phase are located in both the cytosol and the mitochondrial matrix. Here, we identify a mitochondrial carrier in *Blastocystis* that can transport glycolytic intermediates, such as dihydroxyacetone phosphate and glyceraldehyde-3-phosphate, across the mitochondrial inner membrane, linking the cytosolic and mitochondrial branches of glycolysis. Comparative analyses with the phylogenetically related human mitochondrial oxoglutarate carrier (SLC25A11) and dicarboxylate carrier (SLC25A10) show that the glycolytic intermediate carrier has lost its ability to transport the canonical substrates malate and oxoglutarate. *Blastocystis* lacks several key components of oxidative phosphorylation required for the generation of mitochondrial ATP, such as complexes III and IV, ATP synthase, and ADP/ATP carriers. The presence of the glycolytic pay-off phase in the mitochondrial matrix generates ATP, which powers energy-requiring processes, such as macromolecular synthesis, as well as NADH, used by mitochondrial complex I to generate a proton motive force to drive the import of proteins and molecules. Given its unique substrate specificity and central role in carbon and energy metabolism, the carrier for glycolytic intermediates identified here represents a specific drug and pesticide target against stramenopile pathogens, which are of great economic importance.

## eLife assessment

This **important** study identifies candidate mitochondrial metabolite carriers in stramenopile protists that may allow these divergent eukaryotes to maintain a compartmentalized glycolytic pathway. This study fills a gap in our understanding of glycolysis evolution and opens avenues for drug design to combat stramenopile parasites. The evidence, based on phylogenetic analysis, thermostability shift assays, and in vitro reconstitution of transport reactions, is **convincing**, albeit lacking direct in vivo confirmation of the physiological function of these candidates.

**eLife digest** All living organisms breakdown food molecules to generate energy for processes, such as growing, reproducing and movement. The series of chemical reactions that breakdown sugars into smaller molecules – known as glycolysis – is so important that it occurs in all life forms, from bacteria to humans.

In higher organisms, such as fungi and animals, these reactions take place in the cytosol, the space surrounding the cell's various compartments. A transport protein then shuttles the end-product of glycolysis – pyruvate – into specialised compartments, known as the mitochondria, where most energy is produced.

However, recently it was discovered that a group of living organisms, called the stramenopiles, have a branched glycolysis in which the enzymes involved in the second half of this process are located in both the cytosol and mitochondrial matrix. But it was not known how the intermediate molecules produced after the first half of glycolysis enter the mitochondria.

To answer this question, Pyrihová et al. searched for transport protein(s) that could link the two halves of the glycolysis pathway. Computational analyses, comparing the genetic sequences of many transport proteins from several different species, revealed a new group found only in stramenopiles. Pyrihová et al. then used microscopy to visualise these new transport proteins – called GIC-1 and GIC-2 – in the parasite *Blastocystis,* which infects the human gut, and observed that they localise to mitochondria.

Further biochemical experiments showed that GIC-1 and GIC-2 can physically bind these intermediate molecules, but only GIC-2 can transport them across membranes. Taken together, these observations suggest that GIC-2 links the two halves of glycolysis in *Blastocystis.*

Further analyses could reveal corresponding transport proteins in other stramenopiles, many of which have devastating effects on agriculture, such as *Phytophthora*, which causes potato blight, or *Saprolegnia*, which causes skin infections in farmed salmon. Since human cells do not have equivalent transporters, they could be new drug targets not only for *Blastocystis,* but for these harmful pathogens as well.

## Introduction

Glycolysis is a crucial metabolic pathway that most organisms use to obtain energy from the oxidation of sugars, which consists of 10 enzymatic reactions in total, divided into two phases. The first 'preparatory' phase consists of five reactions, the products of which are the triose phosphates dihydroxyacetone phosphate and glyceraldehyde-3-phosphate. The second 'pay-off' phase leads to the production of ATP, NADH, and pyruvate, which is transported into the mitochondrial matrix by the mitochondrial pyruvate carrier (*Bricker et al., 2012*; *Herzig et al., 2012*). In most eukaryotes, glycolysis is a cytosolic process with the exception of trypanosomes, where it is localized in the glycosomes (*Opperdoes and Borst, 1977*). However, it was recently discovered that a large group of eukaryotes, the stramenopiles, have a branched glycolytic pathway in which the pay-off phase is duplicated and also localized in mitochondria (*Abrahamian et al., 2017*; *Río Bártulos et al., 2018*). The stramenopiles are an extremely diverse eukaryotic group consisting mainly of protist algae, which are among the most important primary oxygen producers in the oceans (*Burki et al., 2020*). The group also includes multicellular algae (kelps) and the economically devastating oomycetes, such as *Phytophthora infestans* (*Jiang and Tyler, 2012*), which causes potato blight, one of the major contributing factors of the Great Famine in Ireland in the late 19th century.

The human gut protozoan *Blastocystis* also belongs to the stramenopiles. Despite more than 1 billion infected individuals worldwide (*Andersen and Stensvold, 2016*), the public health significance of *Blastocystis* is debated. *Blastocystis* is often linked to patients with diarrhoea, irritable bowel syndrome (*Poirier et al., 2012*), or inflammatory bowel disease (*Dogruman-Al et al., 2009*). However, it is also frequently found in healthy individuals, and a few microbiome studies associated *Blastocystis* with higher microbial diversity (*Andersen et al., 2015*; *Audebert et al., 2016*; *Stensvold et al., 2022*). The controversy may arise from the extreme genetic diversity of *Blastocystis*, which is classified into over 40 subtypes (*Andersen et al., 2015*; *Stensvold et al., 2023*), and thus its influence on gut microbiota is likely to be subtype specific (*Tito et al., 2019*). Nevertheless, *Blastocystis* ST7-B,

which was used in this study, is most frequently associated with pathogenic phenotypes (*Ajjampur et al., 2016*; *Deng et al., 2023*; *Stensvold et al., 2009*; *Wu et al., 2014*; *Yason et al., 2019*) and was recently linked to a decreased microbial diversity in diarrheal patients (*Deng et al., 2022*). The treatment of choice is metronidazole (*Stenzel and Boreham, 1996*), but there have been reports of resistance (*Haresh et al., 1999*; *Mirza et al., 2011*; *Moghaddam et al., 2005*), highlighting the need for new therapeutic interventions.

Unlike other stramenopiles, *Blastocystis* has lost the cytosolic pay-off phase of glycolysis, consistent with the absence of a gene encoding a mitochondrial pyruvate carrier (*Río Bártulos et al., 2018*). The only exception is the last enzyme of the pathway, pyruvate kinase, which was also identified as a cytosolic protein (*Lantsman et al., 2008*). *Blastocystis* is an anaerobe (*Zierdt, 1991*) with highly divergent mitochondria (*Stechmann et al., 2008*). It has respiratory complexes I and II, but lacks the rest of the electron transport chain, as well as ATP synthase and mitochondrial ADP/ATP carriers (*Gentekaki et al., 2017*; *Stechmann et al., 2008*; *Wawrzyniak et al., 2008*). Therefore, *Blastocystis* relies mainly on glycolysis and fermentation for its metabolic energy requirements (*Müller et al., 2012*). The transport of glycolytic intermediates across the mitochondrial inner membrane is necessary for the completion of glycolysis and is critical for the survival of the parasite. Glycolytic enzymes are powerful drug targets in parasites that are strictly dependent on glycolysis for ATP production. The structural differences between host and parasitic glyceraldehyde-3-phosphate dehydrogenase and triosephosphate isomerase in *Plasmodium falciparum* have been studied with the aim of inhibiting glycolysis (*Bruno et al., 2014*; *Penna-Coutinho et al., 2011*; *Ravindra and Balaram, 2005*). Similarly, inhibitors against phosphofructokinase in *Trypanosoma brucei* have a rapid effect on the parasite viability without inhibiting the host enzyme (*McNae et al., 2021*). Understanding the unique aspects of parasite metabolism is crucial for the discovery of new drug targets.

It is clear that the branched stramenopile glycolysis needs to be linked by a specific transporter in the mitochondrial inner membrane, as the solute exchange between the cytosol and mitochondrial matrix is tightly controlled (*Kunji et al., 2020*; *Ruprecht and Kunji, 2020*). The mitochondrial carrier family (SLC25) is the largest solute transporter family in humans (*Kunji et al., 2020*). Its members facilitate the transport of nucleotides, amino acids, inorganic ions, keto acids, vitamins, and other nutrients across the mitochondrial inner membrane. All SLC25 members have a threefold pseudo-symmetric structure, evident at the sequence level by the presence of three conserved homologous repeats (*Saraste and Walker, 1982*) and specific motifs important for their transport mechanism (*Ruprecht et al., 2019*; *Ruprecht and Kunji, 2020*; *Ruprecht and Kunji, 2021*).

Here, we report a stramenopile-specific group of transport proteins closely related to mitochondrial carboxylate carriers of the SLC25 family. We characterized one of these unique *Blastocystis* carriers, which has lost its ability to transport the canonical substrates malate and oxoglutarate. Instead, this carrier can transport several glycolytic intermediates, including dihydroxyacetone phosphate, glyceraldehyde-3-phosphate, 3-phosphoglycerate, and phosphoenolpyruvate (PEP), thus providing a previously undiscovered transport link between cytosolic and mitochondrial glycolysis. Since *Blastocystis* relies solely on mitochondrial glycolysis, the inhibition of this carrier might represent a promising way to treat this parasite.

## Results

### A group of mitochondrial carboxylate carriers that is unique to stramenopiles

The last enzyme of the cytosolic phase of glycolysis in *Blastocystis* is triosephosphate isomerase (*Río Bártulos et al., 2018*). Consequently, dihydroxyacetone phosphate and glyceraldehyde-3-phosphate need to be transported into the mitochondrial matrix for the pay-off phase of glycolysis. Given the presence of mitochondrial glycolysis in stramenopiles, we searched for a transporter that is not only ubiquitous in stramenopiles, but also absent in other eukaryotes. As glycolysis in *Blastocystis* would be critically dependent on this transporter, we used this parasite as a model organism. We focussed on mitochondrial carboxylate carriers because of their ability to transport phosphorylated three-carbon substrates (*Castegna et al., 2010*; *Fiermonte et al., 2001*; *Palmieri et al., 1972*) and compared the sequences of carboxylate carriers from all major eukaryotic supergroups. Our analyses revealed four well-defined groups of carriers: dicarboxylate carriers (DIC, SLC25A10), oxoglutarate/malate

carriers (OGC, SLC25A11), di/tricarboxylate carriers (DTC), and a new stramenopile-specific group, the putative glycolytic intermediate carriers (GIC) (*Figure 1*). At least one copy of the putative GIC carrier was identified in each stramenopile species. In *Blastocystis* ST7-B, there are four copies of the putative GIC protein, but two of them are truncated and lack crucial mitochondrial carrier elements (*Ruprecht et al., 2019*; *Ruprecht and Kunji, 2020*; *Ruprecht and Kunji, 2021*), possibly rendering these proteins non-functional (XP_012899519.1 has an N-terminal truncation; XP_012896318.1 has a C-terminal truncation) (*Ruprecht and Kunji, 2020*).

## The unidentified *Blastocystis* carriers localize to mitochondria

Another important criterion is that the putative GIC localize to mitochondria in *Blastocystis*. Specific (peptide) rabbit antibodies were raised against the two putative GIC bGIC-1 and bGIC-2 as well as an orthologue of the *Blastocystis* oxoglutarate carrier bOGC (*Figure 2B*). Mitochondria were visualized using antibodies raised against a triosephosphate isomerase/glyceraldehyde-3-phosphate dehydrogenase mitochondrial fusion protein (TPI-GAPDH) (*Río Bártulos et al., 2018*). Immunoconfocal microscopy clearly shows that bOGC, bGIC-1, and bGIC-2 localize to mitochondria in *Blastocystis*, showing that they are potential candidates for the missing transport links (*Figure 2A*).

## Human and *Blastocystis* carriers purified from yeast mitochondria are folded

All three *Blastocystis* proteins were expressed in and purified from *Saccharomyces cerevisiae* mitochondria using the detergent lauryl maltose neopentyl glycol (*Figure 3A*). They were compared to the human oxoglutarate carrier SLC25A11 (hOGC) and dicarboxylate carrier SLC25A10 (hDIC), prepared in the same way. Next, thermostability analysis was used to assess whether the purified proteins are correctly folded. Spectral properties of tryptophan/tyrosine residues change as their local environment changes during protein denaturation in a temperature ramp (*Alexander et al., 2014*). The assay produces an apparent melting temperature (Tm) at which the rate of protein unfolding is the highest. All three proteins produced unfolding curves, showing that they are folded. The apparent melting temperatures for bOGC (52.6 ± 0.5°C), bGIC-1 (60.7 ± 0.4°C), and bGIC-2 (59.8 ± 0.4°C) (*Figure 3B, C*) are similar to those observed for hOGC (51.0°C), hDIC (54.1°C), and other mitochondrial carriers (*Crichton et al., 2015*; *Jaiquel Baron et al., 2021*; *Majd et al., 2018*).

## Two mitochondrial carriers of *Blastocystis* bind glycolytic intermediates

We have previously shown that inhibitors (*Crichton et al., 2015*; *Jaiquel Baron et al., 2021*; *Tavoulari et al., 2022*) and substrates (*Majd et al., 2018*; *Mavridou et al., 2022*) can specifically stabilize a population of transport proteins in thermal denaturation assays. This technique provides a simple, high-throughput method for screening libraries of compounds to identify potential protein binders that can then be confirmed in low-throughput transport assays with radiolabelled substrates (*Majd et al., 2018*). We used detergent-solubilized, purified protein to screen a library of 39 commercially available compounds, which included mono-, di-, and tricarboxylates, acids, inorganic ions, and several glycolytic intermediates (with the exception of 1,3-bisphosphoglycerate and 2-phosphoglycerate, which were not available).

In order to quantify the binding effect of a compound, we calculated a thermostability shift (ΔTm) by subtracting the apparent melting temperature in the absence of a compound from that in its presence. A positive shift suggests that the compound is stabilizing, indicating a putative substrate (*Majd et al., 2018*). We first tested the library on purified hOGC (SLC25A11) and hDIC (SLC25A10). A range of compounds had stabilizing effects on hOGC in thermostability shift assays (*Figure 4A*). Oxoglutarate (3.8 ± 0.4°C), malate (3.6 ± 0.3°C), maleate (3.1 ± 0.1°C), succinate (1.9 ± 0.3°C), and malonate (1.3 ± 0.2°C) all produced positive shifts. For hDIC, larger shifts than for hOGC were observed for malate (4.8 ± 0.2°C), maleate (8.0 ± 0.1°C), succinate (3.0 ± 0.3°C), and malonate (7.3 ± 0.1°C) (*Figure 4B*). Shifts were also observed for sulphate (3.0 ± 0.2°C), thiosulphate (4.0 ± 0.4°C), phosphate (3.2 ± 0.2°C), dihydroxyacetone phosphate (2.5 ± 0.4°C), and glyceraldehyde-3-phosphate (5.6 ± 0.6°C) (*Figure 4B*), which were not observed for hOGC and may reflect a difference in substrate specificity (*Crompton et al., 1974*).

The thermostability shifts for the *Blastocystis* orthologue bOGC were similar to those of hOGC (SLC25A11), although some compounds, most notably citrate and isocitrate, produced small

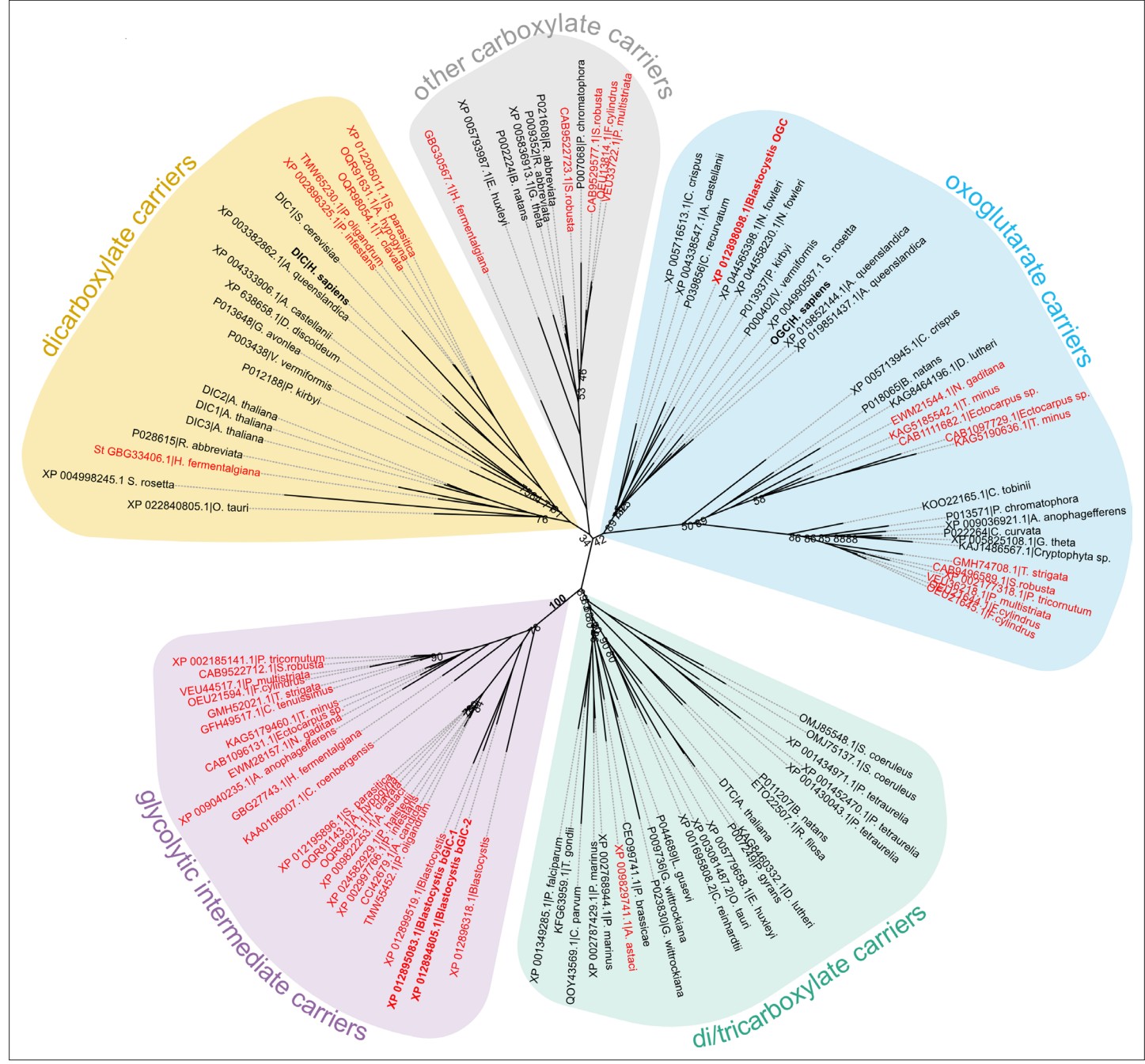

**Figure 1.** Phylogenetic analysis of carboxylate carriers. The maximum likelihood tree was generated by IQ-TREE with the LG+F+R7 model suggested by ModelFinder. The support was calculated using ultrafast bootstrap analysis. The unrooted phylogenetic tree shows the uniqueness of the identified putative stramenopile glycolytic intermediate carriers (GIC) among eukaryotes. The stramenopile-specific GIC (pink) are most closely related to di/tricarboxylate carriers (DTC – green), and to a lesser extend to oxoglutarate carriers (OGC – blue) and dicarboxylate carriers (DIC – yellow). Stramenopile proteins are in red, proteins included in this study are in bold. Note the absence of any non-stramenopile sequences in this clade. Bootstrap values less than 90 and a value for the GIC group are indicated.

The online version of this article includes the following figure supplement(s) for figure 1:

**Figure supplement 1.** The maximum likelihood tree was generated by IQ-TREE with the LG+F+R9 model suggested by ModelFinder.

stabilizations that were not observed for the human orthologue (**Figure 4C**). A wider range of compounds caused an increase in stability of the two candidate carriers bGIC-1 and bGIC-2 compared to bOGC (**Figure 4D, E**). These compounds included glycolytic intermediates, such as glyceraldehyde-3-phosphate (1.0 ± 0.1 and 2.1 ± 0.1°C for bGIC-1 and bGIC-2, respectively) and dihydroxyacetone

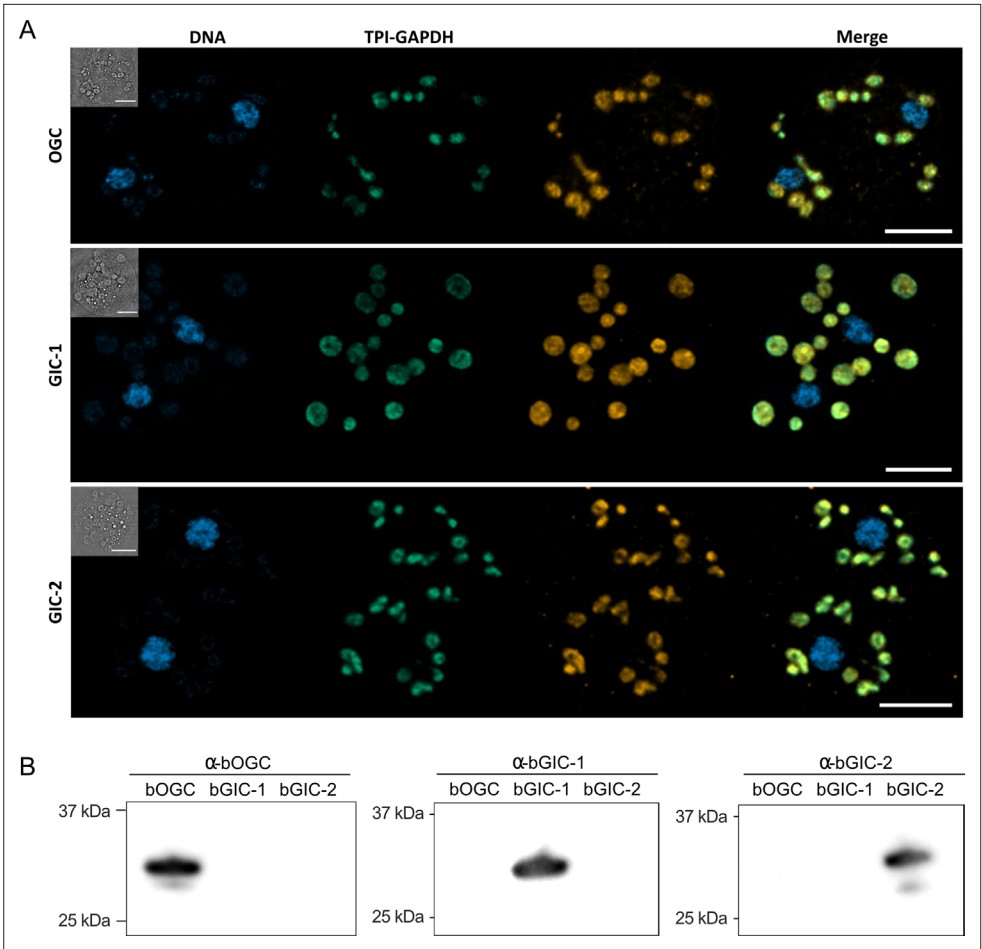

**Figure 2.** Specific antibodies localise glycolytic intermediate carriers in the mitochondria of *Blastocystis*.
(**A**) Localization of glycolytic intermediate carriers (GIC) in *Blastocystis* ST7-B cells using specific antibodies
(orange). The signal colocalizes with the mitochondrial protein TPI-GAPDH (green) (*Río Bártulos et al.,
2018*), demonstrating its mitochondrial localization. DNA is stained with Hoechst 33342 (blue). Inserts top left:
corresponding brightfield image. The scale bar is 5 μm. (**B**) Western blots of *Blastocystis* OGC, GIC-1, and GIC-2
antibodies against purified protein confirm that there is no cross reactivity.

The online version of this article includes the following source data for figure 2:

**Source data 1.** Original file for the western blot analysis in *Figure 2B*.

**Source data 2.** PDF containing *Figure 2B* and original scans of the relevant western blot analysis with highlighted
bands and sample labels.

phosphate (3.6 ± 0.2 and 1.6 ± 0.1 °C) and inorganic ions, such as phosphate (1.7 ± 0.1 and 5.0 ±
0.1°C) and sulphate (2.3 ± 0.2 and 2.6 ± 0.3°C). A significant shift was observed for PEP in the case of
bGIC-2 (4.3 ± 0.2°C). In comparison, hOGC showed no shift for any of these compounds (*Figure 4A*),
whereas bOGC showed only a minor shift in the presence of sulphate (0.5 ± 0.1°C) (*Figure 4C*).

### *Blastocystis* GIC-2 is a functional mitochondrial carrier

We first tested [$^{14}$C]-malate/malate homo-exchange for bGIC-1 and bGIC-2 (*Figure 5A, B*, respec-
tively). No transport was measured, showing that these proteins do not have the ability to transport
malate. We next aimed to find a radiolabelled substrate that is transported by bGIC-1 and bGIC-2. We
tested [$^{33}$P]-phosphate/phosphate and [$^{35}$S]-sulphate/sulphate homo-exchange, as both compounds
induce relatively large shifts in thermostability assays: bGIC-1 (1.7 ± 0.1 and 2.3 ± 0.2°C) and bGIC-2
(5.0 ± 0.1 and 2.6 ± 0.3°C), for phosphate and sulphate, respectively (*Figure 4D, E*). No transport was
detected for either phosphate (*Figure 5C*) or sulphate (*Figure 5E*) for bGIC-1. In contrast, bGIC-2
showed transport activity for both: the rate of [$^{33}$P]-phosphate/phosphate homo-exchange was 4.5

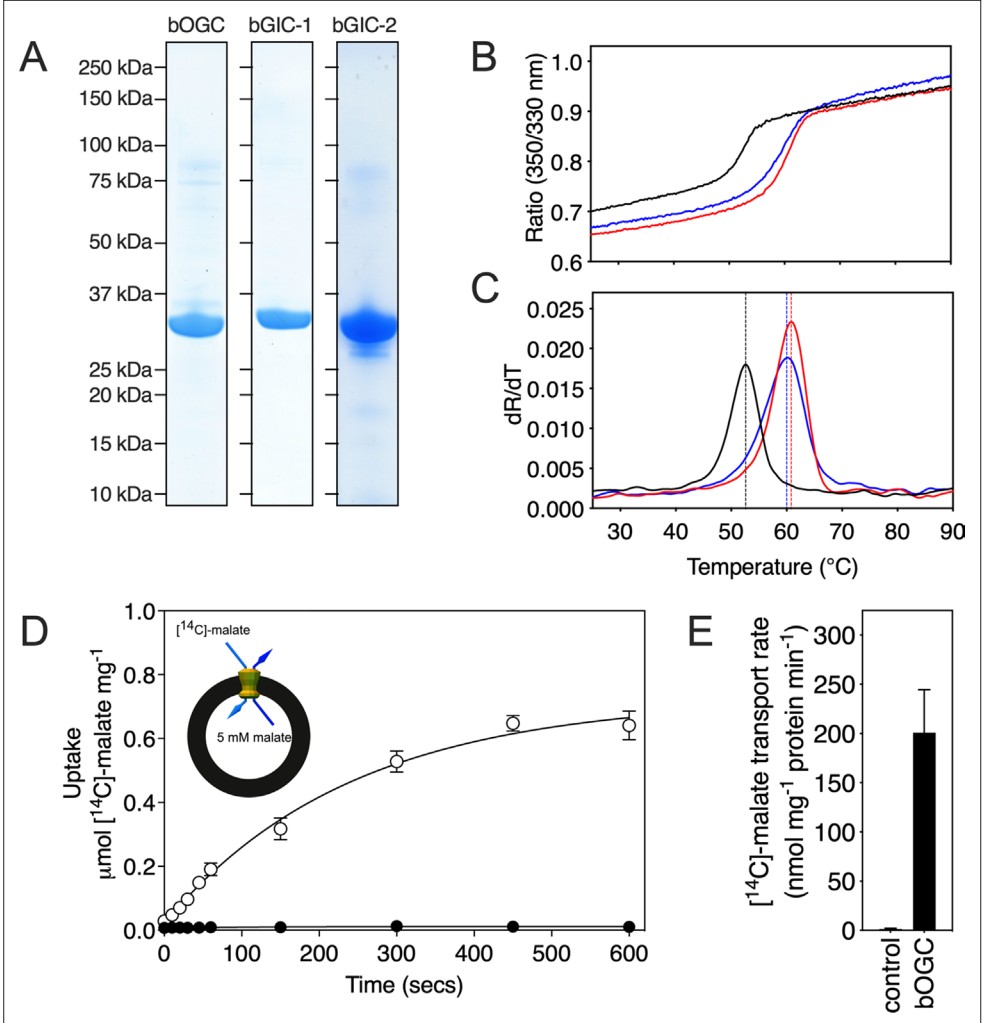

**Figure 3.** Expression, purification, and biophysical characterization of *Blastocystis* carriers. (**A**) Instant-blue stained sodium dodecyl sulphate–polyacrylamide gel electrophoresis (SDS–PAGE) gel of purified bOGC, bGIC-1, and bGIC-2. (**B**) Typical unfolding curves of bOGC (black trace), bGIC-1 (red trace), and bGIC-2 (blue trace) using nanoDSF. (**C**) The peak in the derivative of the unfolding curve (d$R$/d$T$) is the apparent melting temperature (Tm). (**D**) [$^{14}$C]-malate uptake curve of bOGC reconstituted into proteoliposomes loaded with (white circles) or without (black circles) 5 mM malate. Transport was initiated by the external addition of 2.5 µM [$^{14}$C]-malate. (**E**) Initial transport rates, calculated from panel (**D**) from the linear part of the uptake curve (first 60 s). The data represent the mean and standard deviation is of $n$ = 4 (two biological repeats, each with two technical repeats).

The online version of this article includes the following source data for figure 3:

**Source data 1.** Original files for the sodium dodecyl sulphate–polyacrylamide gel electrophoresis (SDS–PAGE) gels shown in *Figure 3A*.

**Source data 2.** PDF containing *Figure 3A* and original scans of the relevant sodium dodecyl sulphate–polyacrylamide gel electrophoresis (SDS–PAGE) gels with highlighted bands and sample labels.

**Source data 3.** Data used for generating graphs in *Figure 3B–E*.

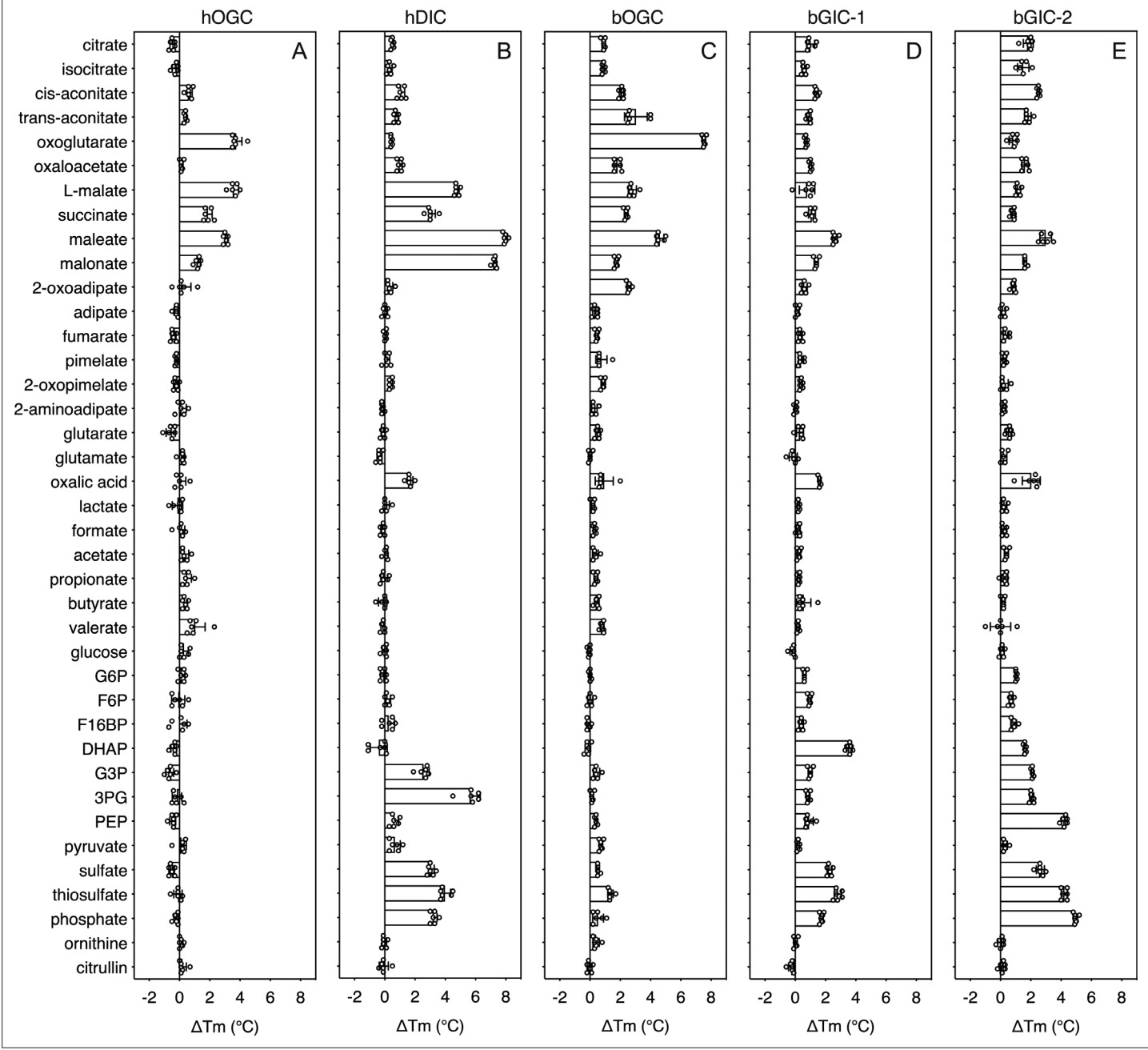

**Figure 4.** Thermostability screening of detergent-solubilized, purified protein using nanoDSF (Prometheus, Nanotemper). (**A**) hOGC, (**B**) hDIC, (**C**) bOGC, (**D**) bGIC-1, and (**E**) bGIC-2 against 10 mM di- and tricarboxylates, glycolytic intermediates, and inorganic ions. The temperature shift (ΔTm) is the apparent melting temperature in the presence of compound minus the apparent melting temperature in the absence of compound. The data are represented by the mean and standard deviation of two biological repeats, each in triplicate (n = 6 in total). G6P, glucose-6-phosphate; F6P, fructose-6-phosphate; F16BP, fructose 1–6 bisphosphate; DHAP, dihydroxyacetone phosphate; G3P, glyceraldehyde-3-phosphate; 3PG, 3-phosphoglycerate; PEP, phosphoenolpyruvate.

The online version of this article includes the following source data for figure 4:

**Source data 1.** Data used for generating graphs in *Figure 4A–E*.

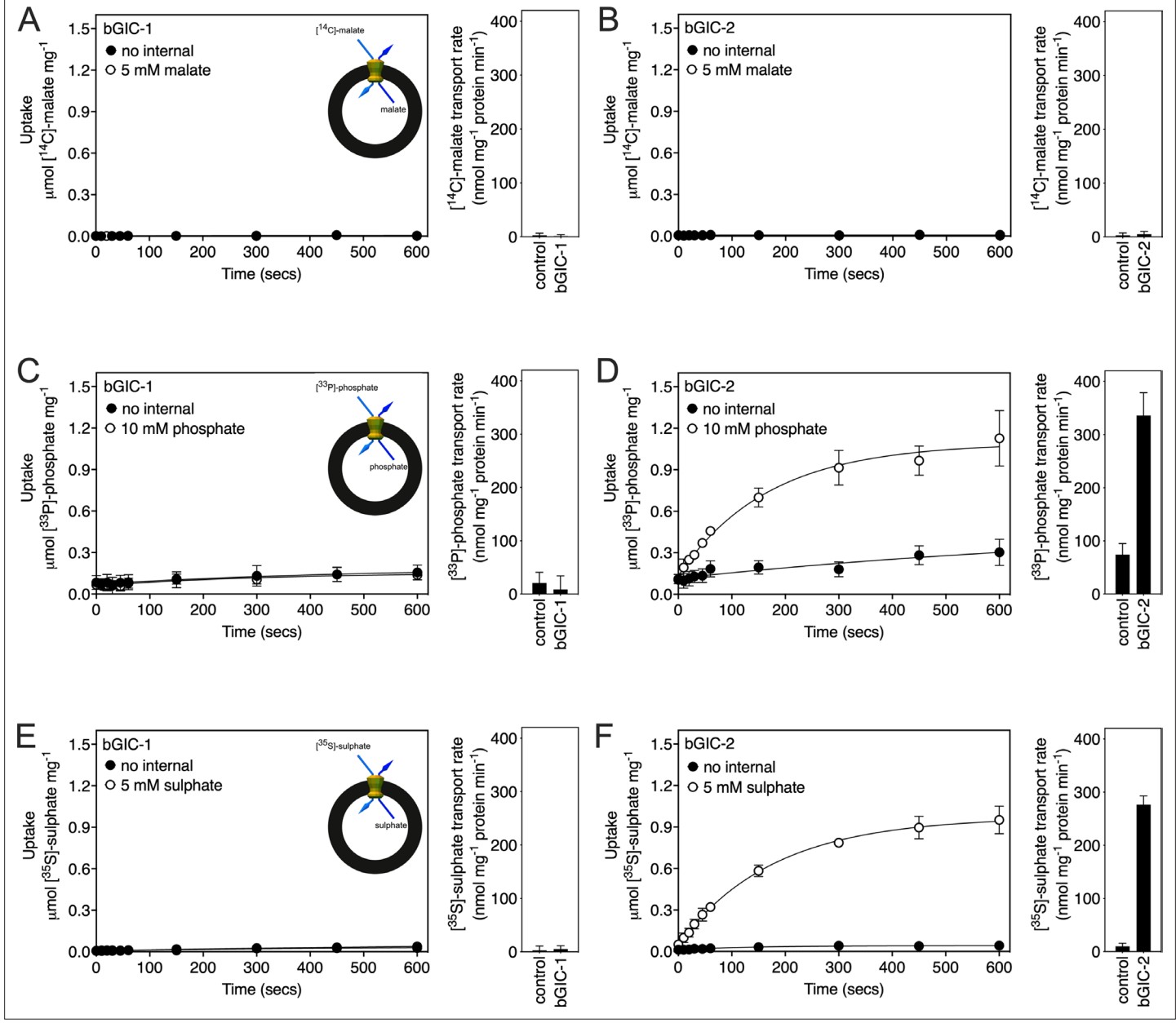

**Figure 5.** Proteoliposomes containing *Blastocystis* putative glycolytic intermediate carriers 1 (bGIC-1; **A, C, E**) and 2 (bGIC-2; **B, D, F**) were loaded with either (**A, B**) 5 mM malate, (**C, D**) 10 mM phosphate, or (**E, F**) 5 mM sodium sulphate, and transport initiated by the external addition of (**A, B**) 2.5 µM [14C]-malate, (**C, D**) 200 µM [33P]-phosphate, or (**E, F**) 25 µM [35S]-sulphate. Controls with no internalized substrate were also tested (black circles). Initial transport rates were calculated from the linear part of the uptake curve (60 s). The data represent the mean and standard deviation of two biological repeats, each with two technical repeats (*n* = 4 in total).

The online version of this article includes the following source data for figure 5:

**Source data 1.** Data used for generating graphs in *Figure 5A–F*.

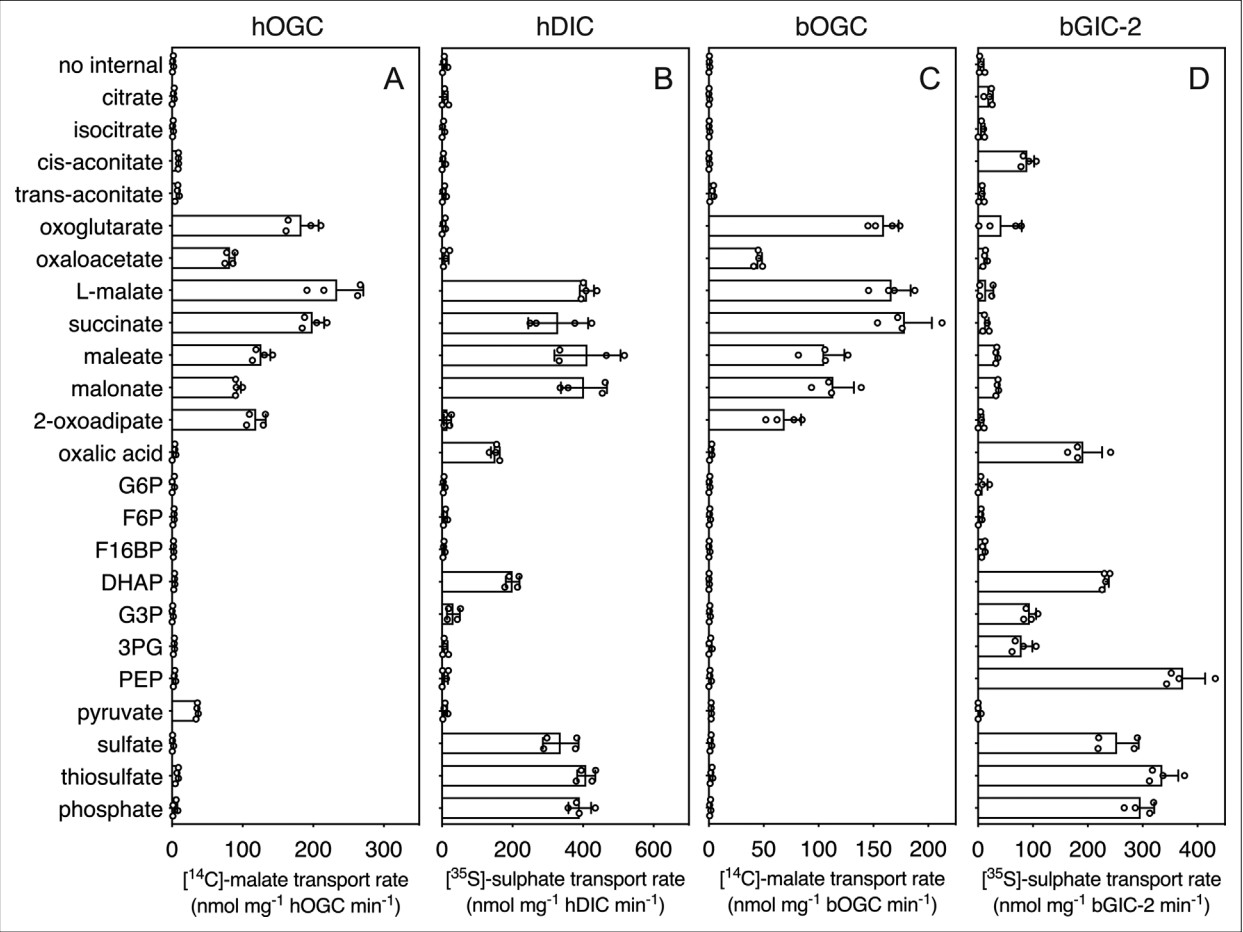

**Figure 6.** Hetero-exchange reactions catalysed by reconstituted carriers. (**A**) Proteoliposomes containing human oxoglutarate carrier (hOGC, SLC25A11), (**B**) human dicarboxylate carrier (hDIC, SLC25A10), (**C**) *Blastocystis* oxoglutarate carrier (bOGC), and (**D**) *Blastocystis* putative glycolytic intermediate carrier 2 (bGIC-2) were loaded with 5 mM compound, and transport initiated by the external addition of 2.5 µM [14C]-malate (hOGC and bOGC) or 25 µM [35S]-sulphate (hDIC and bGIC-2). Initial transport rates were calculated from the linear part of the uptake curve (60 s). The data represent the mean and standard deviation of two biological repeats, each with two technical repeats (*n* = 4 in total). G6P, glucose-6-phosphate; F6P, fructose-6-phosphate; F16BP, fructose 1–6 bisphosphate; DHAP, dihydroxyacetone phosphate; G3P, glyceraldehyde-3-phosphate; 3PG, 3-phosphoglycerate; PEP, phosphoenolpyruvate.

The online version of this article includes the following source data for figure 6:

**Source data 1.** Data used for generating graphs in *Figure 6A–D*.

times above the background (*Figure 5D*), and the rate of [35S]-sulphate/sulphate homo-exchange was 22 times above background (*Figure 5F*). These data show that bGIC-2 is a functional mitochondrial carrier.

## Human OGC and DIC have distinct and overlapping substrate specificities

In addition to the glycolytic intermediates that were the focus of this study, we selected all compounds that stabilized the proteins in thermostability assays and performed transport assays. To test whether a compound was transported, we loaded proteoliposomes with substrate by freeze–thaw extrusion (*Jaiquel Baron et al., 2021*), and initiated exchange with the addition of radiolabelled [14C]-malate (hOGC and bOGC, *Figure 3D, E*) or [35S]-sulphate (hDIC). Only when the internalized compound is a transported substrate of the carrier will the uptake of radiolabelled substrate occur. For hOGC (SLC25A11), uptake was measured for oxoglutarate, oxaloacetate, malate, succinate, maleate, malonate, and 2-oxoadipate (*Figure 6A*). Surprisingly, a low uptake rate was measured for pyruvate, which did not induce a significant shift (0.1 ± 0.3°C). Pyruvate is a small molecule and therefore

probably does not introduce enough additional bonds to stabilize the protein. The substrate specificity of hOGC is similar to that of the rat and bovine oxoglutarate carrier (*Bisaccia et al., 1988*; *Fiermonte et al., 1993*; *Palmieri et al., 1972*). For hDIC (SLC25A10), uptake was observed for malate, succinate, maleate, malonate, sulphate, thiosulphate, phosphate and, to a lesser extent, oxalic acid and dihydroxyacetone phosphate (*Figure 6B*). This result is consistent with previous data that have shown that the rat dicarboxylate carrier also transports sulphate and thiosulphate in addition to dicarboxylates (*Crompton et al., 1974*; *Saris, 1980*). Although we measured transport for dihydroxyacetone phosphate, we have not found any evidence for the presence of dihydroxyacetone phosphate inside mitochondria in the literature. It is possible that it is not transported under physiological conditions in competition with dicarboxylates or other substrates. The substrate specificity of *Blastocystis* OGC (*Figure 6C*) is very similar to human OGC (*Figure 6A*), consistent with their close phylogenetic relationship.

## *Blastocystis* GIC-2 links cytosolic and mitochondrial glycolysis

We used [$^{35}$S]-sulphate as a counter-substrate and tested hetero-exchange to determine the substrate specificity of bGIC-2 (*Figure 6D*). No transport was observed for the canonical oxoglutarate carrier substrates, including oxaloacetate, malate, succinate, and 2-oxoadipate; oxoglutarate, maleate, and malonate displayed only low transport rates. Interestingly, bGIC-2 displayed high transport rates for phosphate, sulphate, and thiosulphate and for the glycolytic intermediates dihydroxyacetone phosphate, glyceraldehyde-3-phosphate, 3-phosphoglycerate, and PEP (*Figure 6D*).

## Discussion

Glycolysis is a textbook example of a well-conserved cytosolic pathway. However, the discovery that glycolysis is branched and also mitochondrial in stramenopiles (*Abrahamian et al., 2017*; *Río Bártulos et al., 2018*; *Nakayama et al., 2012*) raises questions about how glycolytic intermediates are transported across the tightly controlled mitochondrial inner membrane. Here, we report the discovery and characterization of a mitochondrial carrier protein unique to stramenopiles that transports the glycolytic intermediates dihydroxyacetone phosphate, glyceraldehyde-3-phosphate, and PEP, providing the heretofore missing transport link.

Since mitochondrial glycolysis is found exclusively in stramenopiles, we searched for a transporter that is ubiquitous in these species, but not in other eukaryotes. Phylogenetic analyses identified a new class of mitochondrial carboxylate carriers in stramenopiles that are most closely related to plant DTC (*Figure 1*). The plant carriers catalyse the transport of tricarboxylates (citrate, isocitrate, and aconitate) in exchange for dicarboxylates (oxoglutarate, oxaloacetate, malate, maleate, succinate, and malonate) (*Picault et al., 2002*) and are also present in Alveolata, Rhizaria, and Haptophyta (*Toleco et al., 2020*), which share a common evolutionary ancestor with stramenopiles (*Burki et al., 2020*). Therefore, the GIC of stramenopiles is probably derived from a common ancestor with DTC and does not exist in any other eukaryote outside of the stramenopile clade.

*Blastocystis* was used in this study as a stramenopile model organism, as it relies solely on glycolysis for its energy demands and does not have a second branch of glycolysis in the cytosol. We characterized human and *Blastocystis* carboxylate carriers and demonstrated that the stramenopile GIC proteins have a unique substrate specificity that is distinct from the human carriers (*Figure 6*). We also characterized the substrate specificity of *Blastocystis* OGC, which has the same specificity as human OGC (SLC25A11), and we demonstrated that it is unable to transport glycolytic intermediates and thus could not be the missing transport step in the branched glycolytic pathway.

*Blastocystis* GIC-1 did not transport any of the tested substrates (*Figure 5A, C, E*), but it is expressed and localized to mitochondria (*Figure 2A*), demonstrating that it is not a pseudogene. We did not test radiolabelled glycolytic intermediates directly in transport, as PEP and dihydroxyacetone phosphate are not available, and radiolabelled glyceraldehyde-3-phosphate is prohibitively expensive. We cannot, therefore, exclude the possibility that GIC-1 catalyses glycolytic intermediate exchange, but not in hetero-exchange reactions with malate, phosphate, and sulphate. It is more likely that bGIC-1 has lost its transport activity and could possibly fulfil a different role. For example, in *Tetrahymena thermophila*, a non-functional OGC-like carrier is incorporated into a respiratory supercomplex (*Zhou et al., 2022*). In contrast to bGIC-1, bGIC-2 transports several inorganic ions and

glycolytic intermediates, including dihydroxyacetone phosphate and glyceraldehyde-3-phosphate (*Figure 6D*), but not the canonical substrates of *Blastocystis* OGC, such as malate, maleate, malonate, and succinate. Most mitochondrial carriers operate as strict antiporters, where the chemical gradient of one substrate drives the uptake of the counter-substrate (*Ruprecht and Kunji, 2020*). A mitochondrial phosphate carrier homologue was identified in the phylogenetic analysis of *Blastocystis* (*Figure 1—figure supplement 1*), which is driven by the proton motive force to maintain high concentrations of phosphate in the mitochondrial matrix, which in turn could drive the uptake of glycolytic intermediates. *Blastocystis* lacks many of the conserved carrier proteins found in the mitochondria of aerobes. In total, we have only identified 18 mitochondrial carriers in *Blastocystis* (*Figure 1—figure supplement 1*). However, we show that bGIC-2 seems to be able to compensate for the loss of some mitochondrial carriers, including the transport of inorganic ions in the absence of a DIC homologue. Additionally, the distantly related citrate carrier, which transports dicarboxylates, tricarboxylates, and PEP, was also lost in *Blastocystis*, possibly due to an incomplete tricarboxylic acid (TCA) cycle devoid of citrate synthase, aconitase, isocitrate dehydrogenase, and α-ketoglutarate dehydrogenase (*Stechmann et al., 2008*; *Zierdt et al., 1988*). In *Blastocystis*, PEP can be synthesized from mitochondrial pyruvate via oxaloacetate (*Gentekaki et al., 2017*; *Stechmann et al., 2008*), and here we show that PEP transport is catalysed by bGIC-2. When PEP is exported by bGIC-2, it could be converted into pyruvate and lactate by cytosolic pyruvate kinase and lactate dehydrogenase, respectively (*Lantsman et al., 2008*), which would lead to the synthesis of ATP and the conversion of NADH to NAD$^+$ in the cytosol. This reaction may play a significant role in cytosolic ATP generation, as there is no mitochondrial ADP/ATP carrier in *Blastocystis* for adenine nucleotide exchange (*Figure 1—figure supplement 1*). *Blastocystis* also contains a glycerol-3-phosphate dehydrogenase (*Gentekaki et al., 2017*), which oxidizes NADH to convert glycerol-3-phosphate to dihydroxyacetone phosphate, which in turn can be transported into mitochondria by bGIC-2 to generate ATP and reducing equivalents. In most other eukaryotes, cytosolic NADH is oxidized by the malate–aspartate shuttle, but this pathway requires an orthologue of the mitochondrial aspartate/glutamate carrier, which could not be identified in *Blastocystis*, even though it might have a mitochondrial glutamate carrier (*Figure 1—figure supplement 1*).

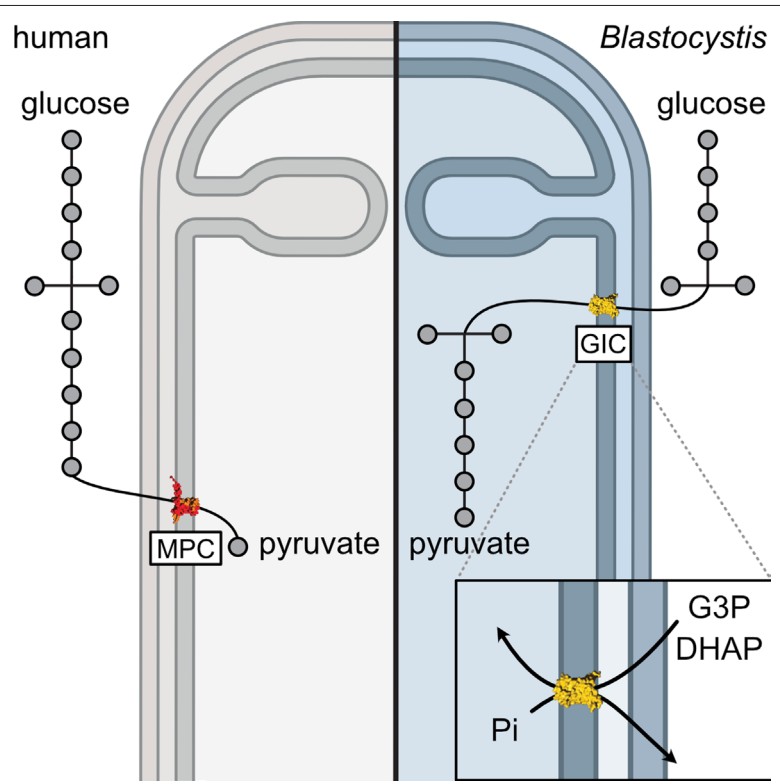

**Figure 7.** Comparing glycolysis in humans and *Blastocystis*.

What selective advantage could there be to have the pay-off phase of glycolysis in the mitochondrial matrix? *Blastocystis* is well adapted to an anoxic environment with its anaerobic metabolism but has a limited ability to detoxify oxygen. In its natural environment, *Blastocystis* likely uses protons as terminal electron acceptors by using pyruvate:ferredoxin oxidoreductase, which would result in the production of hydrogen and a range of fermentative products, such as lactate, acetate, and propionate (*Gentekaki et al., 2017*; *Lantsman et al., 2008*; *Stechmann et al., 2008*). Under aerobic stress, such as in a dysbiotic gut (*Stensvold and van der Giezen, 2018*), it possibly uses oxygen as the terminal electron acceptor by using alternative oxidase, which produces water (*Tsaousis et al., 2018*). However, as a likely adaptation to the anoxic environment, *Blastocystis* has lost some of the canonical components of oxidative phosphorylation, including complexes III and IV of the respiratory chain, ATP-synthase and mitochondrial ADP/ATP carriers (*Stechmann et al., 2008*). Consequently, it has lost its ability to synthesize mitochondrial ATP via this route. This role was taken over by transfer of the pay-off phase of glycolysis to the mitochondrial matrix in *Blastocystis*, which has several advantages. The degradation of one glucose molecule in the cytosol requires one ATP and one pyrophosphate molecule, but will generate four ATP molecules in the mitochondrial matrix, which can be used for ATP-requiring processes, such as protein import, mitochondrial DNA replication and transcription, and mitochondrial protein synthesis (*Denoeud et al., 2011*; *Gentekaki et al., 2017*). In addition, two $NAD^+$ molecules are reduced to NADH, which can be used by complex I to generate a proton motive force to drive the import of proteins and the uptake of ions and molecules into mitochondria, such as phosphate. Cytosolic ATP can be supplied either through the export of PEP by bGIC-2, as explained, or via a mitochondrial carrier that shares properties with the mitochondrial Mg-ATP/Pi carrier, which is generally used to equilibrate adenine nucleotide pools between the matrix and cytosol.

Our results show that the canonical metabolic pathways studied in common model organisms can be altered in non-model eukaryotic species (*Figure 7*). Moreover, the newly discovered glycolytic intermediates carrier demonstrates the plasticity of mitochondrial carriers in terms of their substrate specificity. Through the discovery of this carrier, we have linked the two halves of glycolysis and provided a potential target for specific treatment not only for *Blastocystis* but also for other stramenopile pathogens that threaten global food security.

## Materials and methods
### In silico analysis

Bioinformatics-based approaches were used to identify potential candidates for a glycolytic intermediate transporter. Any potential transporter should be present in all stramenopiles, but absent in other eukaryotes, and could plausibly transport phosphorylated three carbon substrates. Most of the transport proteins in mitochondria belong to the mitochondrial carrier family (SLC25) (*Kunji et al., 2020*; *Ruprecht and Kunji, 2020*), and thus the initial search focussed on them.

Mitochondrial carrier sequences were gathered using 53 human mitochondrial carriers as a BLAST query against the genomes of selected stramenopiles: *Blastocystis ST7-B*, *Aureococcus anophageferens*, *Ectocarpus* sp., *Saprolegnia parasitica*, *Phaeodactylum tricornutum*, and *Phytopthora infestans*. InterProScan (https://www.ebi.ac.uk/interpro/search/sequence) was then used to discard non-carrier hits (*Jones et al., 2014*), and partial sequences were removed (sequences with fewer than three signature repeats or 250 amino acids). Reciprocal BLAST was then used to eliminate incorrectly assigned sequences. The stramenopile sequences were aligned to *Homo sapiens, S. cerevisiae*, and *Arabidopsis thaliana* carriers. To prepare the alignment for phylogenetic analysis, the structures of the ADP/ATP carrier (PDB: 1OKC and 6GCI) were used to identify and remove the hypervariable loop regions in each sequence.

IQ-TREE was used to calculate a phylogenetic tree using an LG+F+R9 model suggested by ModelFinder and branch supports were generated by ultrafast bootstrap with 1000 replicates (*Hoang et al., 2018*; *Minh et al., 2020*). A clade of stramenopile-specific transporters was identified as a sister group to the *A. thaliana* DTC homologues (*Figure 1—figure supplement 1*). Within this clade, all selected stramenopile species were represented at least once. Interestingly, *Blastocystis*, which has no homologue of the mitochondrial pyruvate carrier, and would have to rely solely on the import of glycolytic intermediates, has four copies of this carrier. Given that DTC substrates are three-carbon

dicarboxylates and tricarboxylates, it is possible that a closely related carrier could transport phosphorylated three-carbon substrates.

To investigate whether this clade is not only present in stramenopiles, but also unique to them, BLAST searches were performed using the unique stramenopile sequences against genomes from an expanded range of eukaryotes using NCBI and EukProt databases (*Priyam et al., 2019*). The collected sequences were screened to remove partial and duplicate sequences and sequences that shared contact points with carboxylate carriers (*Kunji and Robinson, 2006*; *Robinson and Kunji, 2006*) were selected. They were added to a multiple sequence alignment of the identified stramenopile carriers and other closely related transporters from the first tree. The maximum likelihood tree was generated by IQ-TREE with the LG+F+R7 model suggested by ModelFinder (*Minh et al., 2020*). Branch supports were generated by ultrafast bootstrap with 1000 replicates (*Hoang et al., 2018*).

To address the absence of GIC in other species and exclude the possibility of its presence outside of stramenopiles, we performed additional analyses. The stramenopile GIC sequences were blasted against NCBI datasets encompassing major eukaryotic groups, and the top hits were selected and aligned with GIC sequences. Phylogenetic analysis of this expanded dataset revealed that the non-stramenopile carriers are more closely related to other carboxylate carriers, indicating that even the carriers most similar to GIC do not fall within the GIC clade. These analyses confirmed that the putative GIC carriers are universal in stramenopiles and also unique to them.

## Protein production in mitochondria of *S. cerevisiae*

Genes (SLC25A11, the human oxoglutarate carrier (hOGC), UniProt accession code: Q02978; SLC25A10, the human dicarboxylate carrier (hDIC), UniProt accession code: Q9UBX3; *Blastocystis* ST7-B oxoglutarate carrier (bOGC), UniProt accession code: D8M7L1; *Blastocystis* ST7-B potential glycolytic intermediate carrier 1 (bGIC-1), UniProt accession code: D8LYZ6; *Blastocystis* ST7-B potential glycolytic intermediate carrier 2 (bGIC-2), UniProt accession code: D8LWW2) were codon-optimized with N-terminal eight histidine tag and a Xa protease cleavage site to aid purification (hOGC, Δ2–13; hDIC, GKP added (positions 2–4); bGIC-1, Δ2–3; bGIC-2, Δ2). The modified genes were cloned into a pYES2/CT vector (Invitrogen) under the control of an inducible galactose promoter, as previously described (*King and Kunji, 2020*), and expressed in *S. cerevisiae* (GenScript) strain W303.1B (hOGC and bGIC-2) or the protease-deficient strain BJ2186 (hDIC, bOGC, and bGIC-1). Transformants were selected on Sc-Ura + 2% (wt/vol) glucose plates. For large-scale expression, a pre-culture of cells grown in Sc-Ura + 2% (wt/vol) glucose was inoculated into 50 l of YPG plus 0.1% glucose medium in an Applikon Pilot Plant 140 l bioreactor. Cells were grown at 30°C for 20 hr, induced with a final concentration of either 0.4% (hOGC, hDIC, and bGIC-1) or 2.0% (bOGC and bGIC-2) galactose, grown for a further 4 (hOGC) or 8 hr (hDIC, bOGC, bGIC-1, and bGIC-2), and harvested by centrifugation (4000 × $g$, 20 min, 4°C). Alternatively, 15 l of culture were grown in shaker flasks and induced as above. Crude mitochondria were prepared using a bead mill (Dyno-Mill Multilab, Willy A. Bachofen AG) by established methods (*King and Kunji, 2020*).

## Preparation of lipid for protein purification

Tetraoleoyl cardiolipin (18:1) powder was purchased from Avanti Polar Lipids. Lipids were solubilized in 10% (wt/vol) lauryl maltose neopentyl glycol (Anatrace) by vortexing for 4 hr at room temperature to give 10 mg ml$^{-1}$ lipid in a 10% (wt/vol) detergent stock. The stocks were stored in liquid nitrogen.

## Protein purification by nickel affinity chromatography

Protein was purified as described previously for the human mitochondrial ADP/ATP carrier (*Jaiquel Baron et al., 2021*). In total, 0.5 g of mitochondria were solubilized in 1.5% (wt/vol) lauryl maltose neopentyl glycol, protease inhibitor tablet (Roche), 20 mM imidazole, and 150 mM NaCl for 1 hr by rotation at 4°C. The insoluble material was separated from the soluble fraction by centrifugation (200,000 × $g$, 45 min, 4°C). Nickel Sepharose slurry (0.5 ml resin; GE Healthcare) was added to the soluble fraction; the mixture was stirred at 4°C for 1 hr. The nickel resin was harvested by centrifugation (100 × $g$, 10 min, 4°C), transferred to a Proteus 1-step batch midi spin column (Generon), and washed with 40 column volumes of buffer A (20 mM HEPES, pH 7.0, 150 mM NaCl, 40 mM imidazole, 0.1 mg ml$^{-1}$ tetraoleoyl cardiolipin/0.1% (wt/vol) lauryl maltose neopentyl glycol) (100 × $g$, 5 min, 4°C); followed by 10 column volumes of buffer B (20 mM HEPES, pH 7.0, 50 mM NaCl, 0.1 mg ml$^{-1}$

tetraoleoyl cardiolipin/0.1% (wt/vol) lauryl maltose neopentyl glycol) (100 × $g$, 5 min, 4°C). The nickel resin was resuspended in buffer B and incubated with 10 mM imidazole, 20 µg factor Xa protease (NEB), and 5 mM $CaCl_2$ with inversion at 10°C overnight. Protein was eluted by centrifugation on a spin column (100 × $g$, 2 min, 4°C). Imidazole and NaCl were removed using a PD-10 desalting column according to the manufacturer's instructions (GE Healthcare) at 4°C and the concentration was determined by spectrometry (NanoDrop Technologies) at 280 nm using calculated extinction coefficients (hOGC: 29,000 $M^{-1}$ $cm^{-1}$; hDIC: 20,900 $M^{-1}$ $cm^{-1}$; bOGC: 50,350 $M^{-1}$ $cm^{-1}$; bGIC-1: 48,735 $M^{-1}$ $cm^{-1}$; bGIC-2: 41,620 $M^{-1}$ $cm^{-1}$).

## Generation and testing of antibodies

Specific peptides (C+LSGEGTSEKLYSSSF for bOGC, C+IVAPGEARLGSIKMA for bGIC-1, and C+TA-AEGRISGMAIAKS for bGIC-2) were generated as a N-terminal Keyhole limpet haemocyanin fusion to raise the antibodies in rabbits (Eurogentec). For western blots, an overnight incubation with 1:2000 α-bOGC, 1:6000 α-bGIC1, and 1:1000 α-bGIC2 and 1-hr incubation with secondary 1:20,000 goat α-rabbit-HRP (Millipore) against 100 ng of purified protein was used. The membranes were developed with ECL Western Blotting Detection Reagents (Amersham).

## Cell culture

Axenic cultures of *Blastocystis* ST7-B were a generous gift from Professor Kevin S. W. Tan (National University of Singapore) and were grown as described previously (*Leonardi et al., 2021*).

## Confocal microscopy

*Blastocystis* ST7-B cells were harvested by centrifugation (1000 × $g$, 10 min) at ambient temperature. Cells were washed with pre-reduced phosphate-buffered saline (PBS), and fixed with 4% methanol-free formaldehyde (Thermo Fisher Scientific), incubated at 37°C for 30 min. Fixed parasites were permeabilized with 0.1% Triton X-100, and blocked with 3% bovine serum albumin (BSA) prior to antibody staining. Rabbit antibodies against bGIC-1, bGIC-2, and bOGC (Eurogentec) were diluted with 3% BSA in PBS with 0.01% Triton X-100 at 1:5000, 1:1000, and 1:1500, respectively, and incubated overnight at 4°C. Guinea pig anti-TPI-GAPDH (1:1000; Eurogentec), previously shown to localize in the mitochondria (*Río Bártulos et al., 2018*) was used as an mitochondrial marker. Alexa Fluor 488-conjugated goat anti-rabbit (A-21428, Invitrogen) and Alexa Fluor 555-conjugated goat anti-guinea pig (A-11073, Invitrogen), used as secondary antibodies, were diluted with 3% BSA in PBS with 0.01% Triton X-100 at 0.5 and 2 µg $ml^{-1}$, respectively. Hoechst 33342 (5 µg $ml^{-1}$, Thermo Fisher Scientific) was used to stain DNA. All immunofluorescence assays were carried out in No. 1.5 high precision glass cover slips (Marienfeld) coated with 20 µg Cell-Tak (Corning). Cells were mounted using ProLong Gold Antifade Mountant (Molecular Probes) and cured overnight. Images were taken using a Leica SP8 laser scanning confocal microscope (Leica Microsystems) equipped with oil immersion objective lens (Leica Microsystems, HC PL APO CS2 ×100/1.40 NA). Z-stacks were collected with a system-optimized sampling compliant to the Nyquist–Shannon sampling theorem and with LAS X LIGHTNING deconvolution-optimized settings (Leica Microsystems). Optical sections from the image stacks were further processed using Fiji Version 1.54d.

## Thermostability analysis

Thermal unfolding analyses were performed with dye-free differential scanning fluorimetry (nanoDSF), which monitors the changes in the spectral properties of tryptophan and tyrosine residues due to changes in their environment caused by unfolding (*Alexander et al., 2014*). The proteins have the following number of tryptophans: hOGC (two); hDIC (one); bOGC (five); bGIC-1 (five); bGIC-2 (four). Approximately 0.6 µg of protein and 10 mM substrate were added into a final volume of 10 µl purification buffer B, and the samples were loaded into nanoDSF-grade standard glass capillaries. The temperature was increased by 4°C every minute from 25 to 95°C, the intrinsic fluorescence was measured in a Prometheus NT.48 nanoDSF device, and the apparent melting temperature (Tm) was calculated with the PR.ThermControl software (NanoTemper Technologies).

## Reconstitution of carrier protein into liposomes

A mix containing *Escherichia coli* polar lipid extract (Avanti Polar Lipids order number: 100600 C), egg L-α-phosphatidylcholine (Avanti Polar Lipids order number: 840051 C), and tetraoleoyl cardiolipin (Avanti Polar Lipids order number: 710335 C) in a 15:5:1 (wt/wt) ratio was dried under a stream of nitrogen: 6.3 mg of total lipids were used per sample (0.625 ml). Lipids were re-hydrated in 20 mM HEPES, pH 7.0, 50 mM NaCl, and 5 mM $MgCl_2$. The detergent pentaethylene glycol monodecyl ether ($C_{10}E_5$) was added to a final concentration of 2.5% (vol/vol), and the lipids were solubilized by vortexing and incubated on ice: the equivalent 30 µg protein was added per sample. The pentaethylene glycol monodecyl was removed by SM-2 bio-beads (Bio-Rad): five additions of bio-beads were made to the master mix every 20 min with inversion at 4°C: four additions of 30 mg, and the final addition of 240 mg per sample. The samples were incubated overnight at 4°C with rotation. Bio-beads were removed by passage through empty micro-bio spin columns (Bio-Rad). The compound was internalized by freeze–thaw extrusion (*Jaiquel Baron et al., 2021*). Compound, to a final concentration of 5 mM (malate and sulphate assays) or 10 mM (phosphate assays) was added, and the sample was subjected to three cycles of freezing in liquid nitrogen for 2 min followed by thawing in a room temperature water bath for 10 min. Proteoliposomes were subsequently extruded by 21 passages through a 0.4-µm filter (Millipore), and the external substrate was removed and exchanged into buffer (20 mM HEPES, pH 7.0 and 50 mM NaCl) using a PD10 desalting column (GE Healthcare) according to the manufacturer's instructions.

## Transport assays

Transport assays were performed using the Hamilton MicroLab Star robot (Hamilton Robotics Ltd). Proteoliposomes (100 µl) were loaded into the wells of a MultiScreenHTS + HA 96-well filter plate (pore size 0.45 µm, Millipore). Uptake of the radiolabelled substrate was initiated by the addition of 100 µl buffer containing 2.5 µM [$^{14}$C]-malate (American Radiolabelled Chemicals), 25 µM [$^{35}$S]-sodium sulphate (Hartmann Analytic), or 200 µM [$^{33}$P]-orthophosphoric acid (Hartmann Analytic). Uptake was stopped after 0, 10, 20, 30, 45, 60, 150, 300, 450, and 600 s by filtration and washing three times with 200 µl ice-cold buffer (20 mM HEPES, pH 7.0, 50 mM NaCl). Plates were dried overnight, 200 µl Micro-Scint-20 (Perkin Elmer) were added, and levels of radioactivity were determined using a TopCount scintillation counter (Perkin Elmer). Initial rates were determined from the linear part of the uptake curve (60 s).

## Sodium dodecyl sulphate–polyacrylamide gel electrophoresis analysis and protein quantification

Protein was visualized by sodium dodecyl sulphate–polyacrylamide gel electrophoresis analysis using 4–12% bis-tris mPAGE gels and stained using InstantBlue (Abcam), according to the manufacturer's instructions. Reconstitution efficiency was determined by comparing the amount of protein incorporated into proteoliposome samples to a standard curve of known amounts of purified protein.

## Acknowledgements

MSK and ACK were supported by grant MC_UU_00028/2 of the UK Medical Research Council to ERSK, and EP and MRT were supported by Norwegian Research Council grant 301170 to MvdG. We thank Dr. Shane Palmer (MRC Mitochondrial Biology Unit) for carrying out large-scale fermentations in this project, Drs. Denis Lacabanne, Sotiria Tavoulari, Camila Cimadamore-Werthein, and Pavel Dolezal for constructive comments on the manuscript, and Associate Professor Kevin SW Tan for the ST7-B *Blastocystis* culture.

## Additional information

### Funding

| Funder | Grant reference number | Author |
|---|---|---|
| Norges Forskningsråd | 301170 | Eva Pyrihová<br>M Rey Toleco<br>Mark van der Giezen |
| Medical Research Council | MC_UU_00028/2 | Martin S King<br>Alannah C King<br>Edmund RS Kunji |

The funders had no role in study design, data collection, and interpretation, or the decision to submit the work for publication.

### Author contributions

Eva Pyrihová, Martin S King, Data curation, Formal analysis, Validation, Investigation, Visualization, Methodology, Writing – original draft, Writing – review and editing; Alannah C King, M Rey Toleco, Data curation, Formal analysis, Validation, Investigation, Visualization, Methodology, Writing – review and editing; Mark van der Giezen, Edmund RS Kunji, Conceptualization, Resources, Data curation, Formal analysis, Supervision, Funding acquisition, Validation, Investigation, Visualization, Methodology, Writing – original draft, Project administration, Writing – review and editing

### Author ORCIDs

Eva Pyrihová ⓘ http://orcid.org/0000-0001-5611-1234
Martin S King ⓘ http://orcid.org/0000-0001-6030-5154
Alannah C King ⓘ http://orcid.org/0000-0003-4100-3657
M Rey Toleco ⓘ https://orcid.org/0009-0002-7888-8479
Mark van der Giezen ⓘ http://orcid.org/0000-0002-1033-1335
Edmund RS Kunji ⓘ http://orcid.org/0000-0002-0610-4500

Reviewer #1 (Public Review): https://doi.org/10.7554/eLife.94187.3.sa1
Reviewer #2 (Public Review): https://doi.org/10.7554/eLife.94187.3.sa2
Reviewer #3 (Public Review): https://doi.org/10.7554/eLife.94187.3.sa3
Author response https://doi.org/10.7554/eLife.94187.3.sa4

## Additional files

### Supplementary files
• MDAR checklist

### Data availability

All data generated or analysed during this study are included in the manuscript and supporting files; source data files have been provided for Figures 2–6.

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
