## [Editor Report · eLife assessment]

This **important** study identifies candidate mitochondrial metabolite carriers in stramenopile protists that may allow these divergent eukaryotes to maintain a compartmentalized glycolytic pathway. This study fills a gap in our understanding of glycolysis evolution and opens avenues for drug design to combat stramenopile parasites. The evidence, based on phylogenetic analysis, thermostability shift assays, and in vitro reconstitution of transport reactions, is **convincing**, albeit lacking direct in vivo confirmation of the physiological function of these candidates.

---

## [Referee Report · Reviewer #1 (Public Review)]

Summary

This study identifies a family of solute transports in the enteric protist, Blastocystis, that may mediate the transport of glycolytic intermediates across the mitochondrial membrane. The study builds on previous observations suggesting that Blastocystis (and other Stramenopiles) are unusual in having a compartmentalized glycolytic pathway with enzymes involved in upper and lower glycolysis being located in the cytosol and mitochondria, respectively. In this study, the authors identified two putative Stamenopile metabolite transporters that are related to plant di/tricarboxylic acid transporters that might mediate the transport of glycolytic intermediates across the mitochondrial membrane. These GIC-transporters were localized to the Blastocystis mitochondrion using specific rabbit antibodies and shown to bind several glycolytic intermediates (including GAP, DHAP and PEP) based on thermostability shift assays. Direct evidence for transport activity was obtained by reconstituting native proteins in proteoliposomes and measuring uptake of 14C-malate or 35S-sulphate against unlabelled substrates. This assay showed that GIC-2 transported DHAP, GAP and PEP. However, significant transport activity was not observed for bGIC-2. Overall, the study provides strong, but not conclusive evidence that bGIC-2 is involved in transporting glycolytic intermediates across the inner membrane of the mitochondria, while the function of GIC-1 remains unclear, despite exhibiting the same metabolite binding properties as bGIC-2 n thermostability assays.

Strengths:

Overall, the findings are of interest in the context of understanding the diversity of core metabolic pathways in evolutionarily diverse eukaryotes, as well as the process by which cytosolic glycolysis evolved in most eukaryotes. The experiments are carefully performed and clearly described.

Weaknesses:

The main weakness of the study is the lack of direct evidence that either bGIC-1 and/or bGIC2 are active in vivo. While it is appreciated that the genetic tools for disrupting GIC genes in Blastocystis are limited/lacking, are there opportunities to ectopically express or delete these genes in other genetically tractable Stamenopiles, such as Phaeodactylum triconuteum?

The authors demonstrate that both bGIC-1 and bGIC-2 are targeted to the mitochondrion, based on immunofluorescence studies. However, the precise localization and topology of these carriers in the inner or outer membrane is not defined. The conclusions of the study would be strengthened if the authors could show that one/both transporters are present in the inner membrane using protease protection experiments following differential solubilization of the outer and inner mitochondrial membranes.

It is not clear why hetero-exchange reactions were not performed for bGIC-1 (only for bGIC-2).

In both their previous study Bartulos et al (2018) and the current study, the authors have shown that Blastocystis express a TPI-GAPDH fusion protein which is located to the mitochondrion. The presence of the TPI domain in the mitochondrial matrix would obviate the need for bGIC-1/2 triose transporters and decrease their value as drug targets. It is noted that Blastocystis still retains some TPI activity in the cytosol, presumably due to expression of a second cytoplasmic isoform, which could account for the presence of the bGIC transporters. However, some discussion on the role of this mitochondrial TPI-GAPDG fusion protein in Blastocystis and other Stramenopiles would be useful.

The summary slide (Fig 7) in the revised manuscript no longer shows PEP being used as a countersolute for the import of G3P and DHAP. Although it complicates the story, the role of PEP as a counter solute should be shown for completeness and also to make sense of some of the statements in the discussion. In particular, as noted by the authors, mitochondrial PEP could be exported back to the cytsol and converted to pyruvate and/or lactate to generate ATP and NAD, although at the expense of ATP synthesis in the mitochondria.

---

## [Referee Report · Reviewer #2 (Public Review)]

In this manuscript, the authors set out to identify transporters that must exist in Stramenophiles due to the fact that the second half of glycolysis appears to be conducted in the mitochondria. They hypothesize that a Stramenophile-specific clade of transporters related to the dicarboxylate carriers are likely the relevant family and then go on to test two proteins from Blastocystis due to the infectious disease relevance of this organism. They show rather convincingly that these two proteins are expressed and are localized to the mitochondria in the native organism. The purified proteins bind to glycolytic intermediates and one of them, GIC-2, transports several glycolytic intermediates in vitro. This is a very solid and well-executed study that clearly demonstrates that bCIC-2 can transport glycolytic intermediates.

(1) The major weakness is that the authors aren't able to show that this protein actually has this function in the native organism. This could be impossible due to the lack of genetic tools in Blastocystis, but it leaves us without absolute confidence that bGIC-2 is the important glycolytic intermediate mitochondrial transporter (or even that it has this function in vivo).

(2) My impression is that the authors under-emphasize the fact that the hDIC also binds (and is stabilized by) glycolytic intermediates (G3P and 3PG). In the opinion of this reviewer, this might change my interpretation about the uniqueness of the bGIC proteins. They act on additional glycolytic intermediates, but it's not unique.

---

## [Referee Report · Reviewer #3 (Public Review)]

Summary:

Unlike most eukaryotes Blastocystis has a branched glycolysis pathway, which is split between the cytoplasm and the mitochondrial matrix. An outstanding question was how the glycolytic intermediates generated in the 'preparatory' phase' are transported into the mitochondrial matrix for the 'pay off' phase. Here, the authors use bioinformatic analysis to identify two candidate solute carrier genes, bGIC-1 and bGIC-2, and use biochemical and biophysical methods to characterise their substrate specificity and transport properties. The authors demonstrate that bGIC-2 can transport dihydroxyacetone phosphate, glyceraldehyde-3-phosphate, 3-phosphoglycerate and phosphoenolpyruvate, establishing this protein as the 'missing link' connecting the two split branches of glycolysis in this branch of single celled eukaryotes. The authors also present their data on bGIC-1, which suggests a role in anion transport and bOGC, which is a close functional homologue of the human oxoglutarate carrier (hOGC, SLC25A11) and human dicarboxylate carrier (hDIC, SLC25A10).

Strengths:

The results are presented in a clear and logical arrangement, which nicely leads the reader through the process of gene identification and subsequent ligand screening and functional reconstitution. The results are compelling and well supported - the thermal stabilisation data is supported by the exchange studies. Caveats, where apparent, are discussed and rational explanations given.

Weaknesses:

The study does not contain any significant weaknesses in my view. I would like to see the authors include the initial rate plots used in the main figures (possibly as insets), so we can observe the data points used for these calculations. It would also have been interesting to include the AlphaFold models for bGIC-1 and bGIC-2 and a discussion/rationalisation for the substrate specificity discussed in the study.

---

## [Author Response]

The following is the authors’ response to the original reviews.

Reviewer #1 (Public Review):

Summary:This study identifies a family of solute transports in the enteric protist, Blastocystis, that may mediate the transport of glycolytic intermediates across the mitochondrial membrane. The study builds on previous observations suggesting that Blastocystis (and other Stramenopiles) are unusual in having a compartmentalized glycolytic pathway with enzymes involved in upper and lower glycolysis being located in the cytosol and mitochondria, respectively. In this study, the authors identified two putative Stamenopile metabolite transporters that are related to plant di/tricarboxylic acid transporters that might mediate the transport of glycolytic intermediates across the mitochondrial membrane. These GIC-transporters were localized to the Blastocystis mitochondrion using specific rabbit antibodies and shown to bind several glycolytic intermediates (including GAP, DHAP, and PEP) based on thermostability shift assays. Direct evidence for transport activity was obtained by reconstituting native proteins in proteoliposomes and measuring the uptake of 14C-malate or 35S-sulphate against unlabelled substrates. This assay showed that GIC-2 transported DHAP, GAP, and PEP. However, significant transport activity was not observed for bGIC-2. Overall, the study provides strong, but not conclusive evidence that bGIC-1 is involved in transporting glycolytic intermediates across the inner membrane of the mitochondria, while the function of GIC-2 remains unclear, despite exhibiting the same metabolite binding properties as bGIC-2 in thermostability assays.Strengths:Overall, the findings are of interest in the context of understanding the diversity of core metabolic pathways in evolutionarily diverse eukaryotes, as well as the process by which cytosolic glycolysis evolved in most eukaryotes. The experiments are carefully performed and clearly described.

We thank the reviewer for their constructive comments. We note that bGIC-2 is the identified glycolytic intermediate transporter, not bGIC-1.

Weaknesses:The main weakness of the study is the lack of direct evidence that either bGIC-1 and/or bGIC2 are active in vivo. While it is appreciated that the genetic tools for disrupting GIC genes in Blastocystis are limited/lacking, are there opportunities to ectopically express or delete these genes in other Stamenopiles, such as Phaeodactylum triconuteum, to demonstrate function in vivo?

Here, we have identified a transport protein, unique to stramenopiles, which is present in mitochondria of Blastocystis and can bind and transport glycolytic intermediates. We agree that it would have been desirable to confirm that they function as glycolytic intermediate transporters in vivo. However, the reviewer is correct in saying that the genetic tools for disrupting GIC genes in Blastocystis in vivo are not available. While the reviewer mentions the possibility of performing these analyses in Phaeodactylum tricornutum, it is important to note that this species possesses aerobic mitochondria and that the pay-off phase of glycolysis is present in both the mitochondrial matrix and the cytosol. Consequently, any data obtained from this species might not be conclusive and would also not be relevant to the glycolytic metabolism in Blastocystis, the subject of this study.

The authors demonstrate that both bGIC-1 and bGIC-2 are targeted to the mitochondrion, based on immunofluorescence studies. However, the precise localization and topology of these carriers in the inner or outer membrane are not defined. The conclusions of the study would be strengthened if the authors could show that one/both transporters are present in the inner membrane using protease protection experiments following differential solubilization of the outer and inner mitochondrial membranes.

The protein is a member of the mitochondrial carrier family, which are extremely hydrophobic membrane proteins. Those with an established transport function are known to localise consistently to the mitochondrial inner membrane, which is impermeable to charged molecules, whereas the outer membrane is porous through VDAC. Furthermore, when the carriers are overproduced in *Saccharomyces cerevisiae*, the protein is found in the enriched mitochondrial fraction, adding further support to the idea that they are localised to the inner membrane, as the outer membrane has a limited surface area.

It is not clear why hetero-exchange reactions were not performed for bGIC-1 (only for bGIC-2).

Unfortunately, bGIC-1 did not display transport activity when tested in [14C]-malate/malate, [35S]-sulphate/sulphate or [33P]-phosphate/phosphate homo-exchange reactions, as shown in Figure 6 (Figure 5 in the revised manuscript). Phosphoenolpyruvate and dihydroxyacetone phosphate are not available in a radiolabelled form and glyceraldehyde-3-phosphate is prohibitively expensive, so we were unable to test glycolytic intermediates directly in homo-exchange reactions. Hetero-exchange reactions, as performed in Figure 5 (Figure 6 in the revised manuscript) for bGIC-2, are conclusive, as accumulation of the radio-labelled substrate inside the proteoliposomes can only occur, when the internal substrate is exported. It seems that Blastocystis has multiple copies, some of which are coding for dysfunctional carriers, being possible pseudo-genes.

The summary slide depicted in Fig 7 is somewhat simplified and inaccurate. First, the authors show that TPI is located in the mitochondria in this study, while in the summary figure, TPI is shown to be present in both the cytosol and mitochondrial matrix. A cytosolic localization for TPI provides a functional rationale for having a triose-P carrier in the inner membrane - however, this is not supported by the data shown here. Second, if bGIC1/2 uses PEP as a counter ion to import GA3P and DHAP into the mitochondrion, as proposed in Fig 7, the lower glycolytic pathway would be effectively truncated at PEP, removing substrate for pyruvate kinase and formation of pyruvate/ATP. Third, the authors suggest that DHAP may have other functions in the mitochondria although these are not shown in the figure.

Figure 7 presents a schematic comparison of the localisation of glycolysis in humans and Blastocystis, specifically focused on the transport steps of either pyruvate (humans) or glycolytic intermediates (Blastocystis) into the mitochondrial matrix. Most of the metabolism of Blastocystis has been inferred from the presence or absence of genes, encoding for particular enzymes, with the exception of the unusual glycolytic pathway. We feel that overcomplicating this schematic figure would detract from the main message of this analysis. Although the transport data show that PEP, another glycolytic intermediate, is transported, we agree with the reviewer that PEP export cannot be rationalised in the context of our current understanding of the metabolism, and we have changed the figure accordingly.

We have not suggested that DHAP has other functions in mitochondria; on line 230, we state that ‘we have not found any evidence for the presence of dihydroxyacetone phosphate inside mitochondria in the literature. It is possible that it is not transported under physiological conditions in competition with dicarboxylates or other substrates.’

**Reviewer #2 (Public Review):**
In this manuscript, the authors set out to identify transporters that must exist in Stramenophiles due to the fact that the second half of glycolysis appears to be conducted in the mitochondria. They hypothesize that a Stramenophile-specific clade of transporters related to the dicarboxylate carriers is likely the relevant family and then go on to test two proteins from Blastocystis due to the infectious disease relevance of this organism. They show rather convincingly that these two proteins are expressed and are localized to the mitochondria in the native organism. The purified proteins bind to glycolytic intermediates and one of them, GIC-2, transports several glycolytic intermediates in vitro. This is a very solid and well-executed study that clearly demonstrates that bCIC-2 can transport glycolytic intermediates.

We thank the reviewer for their positive comments on the manuscript, and their careful analyses of the presented data.

(1) The major weakness is that the authors aren't able to show that this protein actually has this function in the native organism. This could be impossible due to the lack of genetic tools in Blastocystis, but it leaves us without absolute confidence that bGIC-2 is the important glycolytic intermediate mitochondrial transporter (or even that it has this function in vivo).

Unfortunately, genetic manipulation in Blastocystis is currently not feasible and thus we cannot conduct a comparative metabolic study with the appropriate controls. The gold standard for identification is to prove the function with purified protein directly, which we have done here by using binding studies and transport assays.

(2) It's atypical that the figures and figure panels don't really follow the order of their citation in the text. It's not a big deal, but mildly annoying to have to skip around in the figures (e.g. Figure 3D-E are described in the same paragraph as Figure 5). In addition, to facilitate the flow and a proper understanding I would encourage a reordering between figures 5D and 6 since Figure 6 is needed to understand the results shown in panel 5D, which may lead to confusion.

We agree with the reviewer and have reordered the figures, switching Figure 5 and 6, which makes the manuscript easier to follow.

(3) My impression is that the authors under-emphasize the fact that the hDIC also binds (and is stabilized by) glycolytic intermediates (G3P and 3PG). In the opinion of this reviewer, this might change the interpretation about the uniqueness of the bGIC proteins. They act on additional glycolytic intermediates, but it's not unique.

The reviewer is correct that hDIC is stabilized by both G3P and 3PG, but neither are transported, as shown in Figure 5B (Figure 6B in the revised manuscript). It is not uncommon for compounds to bind to some extend without being transported, as they share certain structural and chemical features with the substrates, which result in stabilisation in thermostability analyses. For example, GTP stabilises the ADP/ATP carrier in thermostability analyses to some extent (Majd et al, 2018), although it is not a transported substrate of the carrier (King et al, 2020). Although thermostability assays are very useful for screening of potential substrates, it is always necessary to carry out transport assays, which are the gold standard for transporter identification.

**Reviewer #3 (Public Review):**
Summary:Unlike most eukaryotes, Blastocystis has a branched glycolysis pathway, which is split between the cytoplasm and the mitochondrial matrix. An outstanding question was how the glycolytic intermediates generated in the 'preparatory' phase' are transported into the mitochondrial matrix for the 'pay off' phase. Here, the authors use bioinformatic analysis to identify two candidate solute carrier genes, bGIC-1, and bGIC-2, and use biochemical and biophysical methods to characterise their substrate specificity and transport properties. The authors demonstrate that bGIC-2 can transport dihydroxyacetone phosphate, glyceraldehyde-3-phosphate, 3-phosphoglycerate, and phosphoenolpyruvate, establishing this protein as the 'missing link' connecting the two split branches of glycolysis in this branch of single-celled eukaryotes. The authors also present their data on bGIC-1, which suggests a role in anion transport and bOGC, which is a close functional homologue of the human oxoglutarate carrier (hOGC, SLC25A11) and human dicarboxylate carrier (hDIC, SLC25A10).Strengths:The results are presented in a clear and logical arrangement, which nicely leads the reader through the process of gene identification and subsequent ligand screening and functional reconstitution. The results are compelling and well supported - the thermal stabilisation data is supported by the exchange studies. Caveats, where apparent, are discussed and rational explanations are given.

We thank the reviewer for their positive and constructive comments on the manuscript.

Weaknesses:The study does not contain any significant weaknesses in my view. I would like to see the authors include the initial rate plots used in the main figures (possibly as insets), so we can observe the data points used for these calculations. It would also have been interesting to include the AlphaFold models for bGIC-1 and bGIC-2 and a discussion/rationalisation for the substrate specificity discussed in the study.

We have shown uptake curves in both Figure 3 and Figure 6 (Figure 5 in the revised manuscript) to provide the typical uptake curves that we record by our robot, and we also show how we calculate the initial rates. We feel that the inclusion of uptake curves for each compound for each carrier (96 uptake curves in total) would make figure 5 (Figure 6 in the revised manuscript) extremely complicated.

It would also have been interesting to include the AlphaFold models for bGIC-1 and bGIC-2 and a discussion/rationalisation for the substrate specificity discussed in the study.

Whilst AlphaFold is an important step forward in the prediction of protein structures, it is not accurate enough at this time to be used for the rationalisation of the substrate specificity. For instance, there are the significant structural differences between the predicted AlphaFold structure of the human uncoupling protein (https://alphafold.ebi.ac.uk/entry/P25874), by and large based on the mitochondrial ADP/ATP carrier, and the experimentally determined structure, especially for the central cavity where the substrate recognition takes place (Jones et al, 2023; Kang & Chen, 2023). More importantly, it is believed that the optimal binding of the substrate takes place in the occluded state (Klingenberg, 2007; Springett et al, 2017), for which we have no structure.

References

Jones SA, Gogoi P, Ruprecht JJ, King MS, Lee Y, Zögg T, Pardon E, Chand D, Steimle S, Copeman DM et al (2023) Structural basis of purine nucleotide inhibition of human uncoupling protein 1. Sci Adv 9: eadh4251

Kang Y, Chen L (2023) Structural basis for the binding of DNP and purine nucleotides onto UCP1. Nature 620: 226-231

King MS, Tavoulari S, Mavridou V, King AC, Mifsud J, Kunji ERS (2020) A single cysteine residue in the translocation pathway of the mitosomal ADP/ATP carrier from Cryptosporidium parvum confers a broad nucleotide specificity. Int J Mol Sci 21: 8971

Klingenberg M (2007) Transport viewed as a catalytic process. Biochimie 89: 1042-1048

Majd H, King MS, Palmer SM, Smith AC, Elbourne LD, Paulsen IT, Sharples D, Henderson PJ, Kunji ER (2018) Screening of candidate substrates and coupling ions of transporters by thermostability shift assays. Elife 7: e38821

Springett R, King MS, Crichton PG, Kunji ERS (2017) Modelling the free energy profile of the mitochondrial ADP/ATP carrier. Biochim Biophys Acta 1858: 906-914